Cell cycle progression in glioblastoma cells is unaffected by pathophysiological levels of hypoxia

Richards Rosalie 1
Jenkinson Michael D. 2
Haylock Brian J. 3
See Violaine 1 violaine@liverpool.ac.uk
1 Institute of Integrative Biology, Department of Biochemistry, University of Liverpool , Liverpool , United Kingdom
2 Institute of Translational Medicine, Clinical Science Centre, University of Liverpool , Liverpool , United Kingdom
3 Department of Clinical Oncology, Clatterbridge Cancer Centre , Bebington , United Kingdom
Piaggio Giulia
Electronic publication date: 2016 Mar 3
Publication date: 2016
Volume: 4
Electronic Location ID: e1755
Received 2016 Jan 14; Accepted 2016 Feb 12
Copyright: ©2016 Richards et al.
Copyright year: 2016
Copyright holder: Richards et al.
License: This is an open access article distributed under the terms of the Creative Commons Attribution License, which permits unrestricted use, distribution, reproduction and adaptation in any medium and for any purpose provided that it is properly attributed. For attribution, the original author(s), title, publication source (PeerJ) and either DOI or URL of the article must be cited.
License URL: https://creativecommons.org/licenses/by/4.0/

Keywords: Hypoxia, Cell cycle, HIF, Glioblastoma, Cell death, Tumour microenvironment, Brain tumours

Funding: Naseem’s Manx Brain Tumour Charity This work was funded by Naseem’s Manx Brain Tumour Charity. The funders had no role in study design, data collection and analysis, decision to publish, or preparation of the manuscript.

==============================
Hypoxia is associated with the increased malignancy of a broad range of solid tumours. While very severe hypoxia has been widely shown to induce cell cycle arrest, the impact of pathophysiological hypoxia on tumour cell proliferation is poorly understood. The aim of this study was to investigate the effect of different oxygen levels on glioblastoma (GBM) cell proliferation and survival. GBM is an extremely aggressive brain tumour with a heterogeneous oxygenation pattern. The effects of a range of oxygen tensions on GBM cell lines and primary cells were assessed using flow cytometry. Results indicate that cell cycle distribution and viability are unaffected by long term exposure (24–96 h) to pathophysiological levels of oxygen (1–8% O2). Both transient cell cycle arrest and small amounts of cell death could only be detected when cells were exposed to severe hypoxia (0.1% O2). No significant changes in p21 protein expression levels were detected. These findings reinforce the importance of using physiologically relevant oxygen tensions when investigating tumour hypoxia, and help to explain how solid tumours can be both hypoxic and highly proliferative, as is the case with GBM.

Introduction

As tumour growth is dependent on the formation of new blood vessels, tumours are often highly vascularised. However, the vasculature is poorly organised and exhibits severe structural and functional abnormalities. This leads to regions of the tumour experiencing a reduced supply of oxygen, known as hypoxia (Folkman, 1990; Vaupel, Kallinowski & Okunieff, 1989; Dewhirst, Cao & Moeller, 2008). Hypoxia is a fundamentally important hallmark of solid tumours and is associated with tumour progression and poor patient prognosis across a broad range of tumour types (Ruan, Song & Ouyang, 2009; Semenza, 2010). Paradoxically, hypoxia has also been reported to induce cell cycle arrest (Box & Demetrick, 2004). This reduction in proliferation could reduce tumour burden, or alternatively it may increase tumour aggressiveness due to the central role of the cell cycle in mediating sensitivity to both chemo- and radiotherapy (Shah & Schwartz, 2001; Pawlik & Keyomarsi, 2004). To resolve the apparent contradiction between the pro-tumorigenic role of hypoxia and its reported anti-proliferative effects, we performed a systematic investigation into the effect of different oxygen levels on glioblastoma (GBM) cell proliferation and survival.

GBM is the most common primary malignant brain tumour with 3 per 100,000 people diagnosed every year (Ostrom et al., 2013). Despite advances in neurosurgery and radiotherapy over the past 40 years, the prognosis remains poor with an average survival time of just 14 months (Anderson et al., 2008). GBM is a diffusely infiltrative Grade IV astrocytic tumour, characterised histopathologically by brisk mitotic activity, cellular and nuclear atypia, vascular thrombosis, microvascular hyperproliferation and necrosis (Reifenberger et al., 2010). As with many solid tumours, GBM is characterised by a heterogeneous pattern of oxygenation. Investigations using oxygen-sensitive electrodes have shown the normal level of oxygen in human brain tissue to be 5–8%. In contrast, mean oxygen levels in high-grade gliomas range from 0.75% to 2.76% (Meixensberger et al., 1993; Kayama et al., 1991; Beppu et al., 2002; Whittle et al., 2010; Collingridge et al., 1999; Rampling et al., 1994), and the larger the hypoxic tumour volume, the poorer the prognosis (Spence et al., 2008).

There are a number of mechanisms through which hypoxia promotes tumour malignancy, including resistance to radio- and chemotherapy (Haar et al., 2012; Harrison & Blackwell, 2004), increased cell migration and invasion (Herrmann et al., 2015), reprogramming towards a cancer stem cell (CSC) phenotype and expansion of CSC populations (Soeda et al., 2009; Fujiwara et al., 2007; Heddleston et al., 2009). At a cellular level, the master regulator of oxygen homeostasis is the transcription factor hypoxia-inducible factor (HIF). HIF promotes cell survival in low oxygen conditions by activating the transcription of genes that regulate processes such as angiogenesis, glycolysis and invasion (Semenza, 2003). HIF-1α expression is correlated with tumour grade in gliomas, with the highest expression found in high-grade gliomas (Zagzag et al., 2000; Sondergaard et al., 2002).

In contrast to other aspects of tumour malignancy, the effects of hypoxia on cell cycle regulation are poorly characterised. It is often stated that hypoxia induces cell cycle arrest, however these observations have been made in severe hypoxia (≤0.1% O2) or anoxia (Box & Demetrick, 2004; Graeber et al., 1994; Amellem & Pettersen, 1991). Investigations using the 2-nitroimidazole EF5, an agent which forms macromolecular adducts in low-oxygen levels as a result of its reductive metabolism (Koch, 2002), have established that the proportion of severely hypoxic cells in brain tumours is low. Rather, the majority of cells are exposed to moderate hypoxia (>0.5% O2) (Evans et al., 2004). Research into the effect of more physiologically relevant oxygen tensions on tumour growth is lacking.

The aim of this study was to investigate the effects of physiological (8% O2), pathophysiological (1% O2) and severe (0.1% O2) levels of hypoxia on GBM cell proliferation and survival. We demonstrate that cell cycle progression in GBM cells is unaffected by pathophysiological levels of hypoxia, and only severe hypoxia is capable of causing transient cell cycle arrest or cell death.

Methods

Cell culture and hypoxic treatment

All reagents were purchased from Life Technologies, unless otherwise stated. U87 cells (ATCC, HTB-14), U251 cells (CLS, 300385) and D566 cells (a kind gift from Professor DD Bigner, Duke University Medical Centre, USA) were maintained in MEM supplemented with 1% sodium-pyruvate and 10% foetal bovine serum (FBS). U251 and D566 cells were supplemented with 1% non-essential amino acids (NEAA). HeLa cells (ECACC, 93021013) were maintained in MEM plus 10% FBS and 1% NEAA. All cells were maintained at 37 °C in 5% CO2. For flow cytometry experiments, 1 × 105 cells were seeded in 6 cm tissue culture dishes (Corning). For hypoxic experiments, cells were incubated in a Don Whitley H35 Hypoxystation (1% O2) or a New Brunswick Galaxy 48R hypoxic incubator (0.1% and 8% O2). A media change was performed after 48 h.

Tumour dissection and primary culture

Samples of primary GBM tumours were received from patients undergoing craniotomy and resection. All patients gave informed written consent to donate their tissue to the Walton Research Tissue Bank, Walton Centre NHS Foundation Trust, which has full approval of the National Research Ethics Service (11/WNo03/2). Primary cell culture was carried out in accordance with the approved guidelines. Tumour samples were transported in MEM plus 1% penicillin-streptomycin (pen-strep), mechanically dissected, and transferred into dissociation medium (10% trypsin 10X and 1% DNase [Sigma] in MEM plus 1% pen-strep). Samples were incubated for 15–30 min at 37 °C and triturated every 5 min. The trypsin reaction was stopped by adding growth medium (MEM plus 20% FBS, 1% sodium-pyruvate and 1% pen-strep). Cells were centrifuged for 5 min at 3,000 rpm, resuspended in growth medium and seeded into a 75 cm2 tissue culture flask.

Flow cytometry

For the viability analyses, adherent cells were washed with phosphate buffered saline (PBS), trypsinised and suspended in Hank’s balanced saline solution (HBSS) together with the floating fraction of cells. The cell suspension was pipetted into a 96 well plate, stained with FITC Annexin V (1:500) and incubated for 15 min in the dark at room temperature (RT). Propidium iodide (PI) was added at a final concentration of 4 µg/mL immediately prior to analyses. Samples were analysed using a Guava EasyCyte Flow Cytometer and cell viability was established using GuavaSoft software (Millipore). For the cell cycle analyses, cells were washed with PBS, trypsinised and resuspended in PBS with 0.1% TritonX-100. PI and ribonuclease A were added at a final concentration of 10 µg/mL. Cells were analysed by flow cytometry and the percentage of cells in G0/G1, S and G2/M phases was established using ModFit LT (Verity Software House).

Western blotting

30–40 µg protein was resolved on a 10% SDS-polyacrylamide gel, transferred onto a nitrocellulose membrane, and probed with primary antibodies against HIF-1α (BD Biosciences 610959, 1:1000), β-actin (Abcam ab8226, 1:1000) and p21 (Santa Cruz sc-396, 1:500) at 4 °C overnight. Membranes were incubated with either an anti-mouse (1:5000) or anti-rabbit (1:3000) horseradish peroxidase-linked secondary antibody (Cell Signalling) for 1 h at RT. Amersham ECL Prime Western Blotting Detection Reagent (GE Healthcare) was used according to the manufacturer’s instructions prior to detection using a G:BOX gel imaging system (Syngene, UK).

Real time reverse transcription polymerase chain reaction (qRT-PCR)

RNA was extracted using a HP RNA isolation kit (Roche) according to the manufacturer’s instructions. Reverse transcription of 1 µg mRNA to cDNA was carried out using a SuperScript VILO cDNA synthesis kit. Real time polymerase chain reaction (qPCR) was performed in triplicate using 10 µl LightCycler® 480 SYBR Green I Master (Roche), 6 µl DNase/RNase-free H2O, 2 µl cDNA and 0.5 µM of each primer. Primers used were as follows: cyclophilin A forward: GCTTTGGGTCCAGGAATGG, reverse: GTTGTCCACAGTCAGCAATGGT; p21 forward: AGCTGCCGAAGTCAGTTCCTT, reverse: GTTCTGACATGGCGCCTCCT; p27 forward: TCCGGCTAACTCTGAGGACA, reverse: GAAGAATCGTCGGTTGCAGG; E2F1 forward: CCGCCATCCAGGAAAAGGTG, reverse: GTGATGTCATAGATGCGCC. qPCR was performed using a LightCycler 480 (Roche); for parameters see Table S1. Results were analysed using LightCycler 480 software (version 1.5.0.39, Roche). The values for target genes were normalised to cyclophillin A and expressed as fold change from control values (20% O2).

Statistical analyses

Statistical significance was determined by ANOVA using SPSS statistical software (IBM).

Results

Pathophysiological hypoxia has minimal effects on GBM cell proliferation and survival, while long-term exposure to severe hypoxia results in cell death

To investigate the effect of different oxygen levels on cell survival, three GBM cell lines (D566, U87 and U251 cells) were exposed to atmospheric (20% O2), physiological (8% O2) and pathophysiological (1% O2) levels of oxygen for up to 96 h. Cell proliferation and survival were measured using Annexin V/PI staining and cell counting using flow cytometry. D566 and U87 cell proliferation was not affected at any time, while U251 cells only displayed a reduction in cell number after 96 h in 1% O2 (Fig. 1A). The viability of all three cell lines was not affected by the different levels of oxygen (Fig. 1B). These results suggest that pathophysiological levels of hypoxia do not significantly affect cell proliferation or survival.

Figure 1 Exposure to physiological and pathophysiological levels of hypoxia has minimal effects on GBM cell proliferation and survival.

GBM cell lines were exposed to 1%, 8% and 20% O2 for the indicated time points. (A) Cell proliferation was assessed by cell counting using flow cytometry. (B) Cell survival was assessed by Annexin/PI staining and flow cytometry. Figure shows mean of three experiments + SEM. *p < 0.05 **p < 0.01.

We investigated whether severe hypoxia (0.1% O2) could have a more pronounced effect on cell fate. In both D566 and U251 cells, a significant reduction in cell number was evident after 48 h in 0.1% O2, while a significant decrease in U87 cell number could only be observed after 96 h (Fig. 2A). A reduction in cell number can be due to a reduction in proliferation or an increase in cell death. The latter was the case for U251 and U87 cells, which showed a significant decrease in viability after 48 h exposure to 0.1% O2 (Fig. 2B). In both U251 cells (Fig. 2C) and U87 cells (Fig. S1), cell death was predominantly due to necrosis. In contrast, the viability of D566 cells was unaffected by exposure to 0.1% O2, suggesting that the reduction in D566 cell number may be due to cell cycle arrest.

Figure 2 Long-term exposure to severe hypoxia causes cell death.

GBM cell lines were exposed to 0.1% and 20% O2 for the indicated time points. (A) Cell proliferation was assessed by cell counting using flow cytometry. (B) Cell survival was assessed by Annexin/PI staining and flow cytometry. (C) Representative example of U251 flow cytometry profile. Figure shows mean of three experiments + SEM. *p < 0.05 **p < 0.01.

Pathophysiological hypoxia has no effect on cell cycle distribution, while severe hypoxia induces a transient cell cycle arrest in D566 cells only

As a more direct assessment of the effect of hypoxia on cell proliferation, cells were exposed to 0.1%, 1% or 20% O2 prior to cell cycle analyses by flow cytometry. There was no difference in cell cycle distribution between cells exposed to 1% O2 and 20% O2 in any of the cell lines tested (Fig. 3A). In U87 and U251 cells there were no significant differences in cell cycle distribution between cells exposed to 0.1% and 20% O2 (Fig. 3A). However, after 48 h in 0.1% O2 D566 cells displayed a 13% increase (p = .002) in the proportion of cells in G1 phase. Interestingly, this cell cycle arrest was only transient and was lost in chronic hypoxia (≥72 h) (Fig. 3A, Fig. S2). To confirm that resistance to hypoxia-induced cell cycle arrest is not an acquired characteristic of cell lines maintained in culture, we conducted the same experiment with primary GBM cells isolated from tumour samples obtained at resection (designated GBM03 and GBM04). We saw no evidence of cell cycle arrest in 1% or 0.1% O2 (Fig. 3B). These results show that only severe hypoxia has the potential to induce cell cycle arrest in GBM cells and, furthermore, this arrest is only transient.

Figure 3 Severe hypoxia causes a transient G1 phase arrest in D566 cells only.

Cells were exposed to 0.1%, 1% and 20% O2 for the indicated time points and cell cycle distribution was assessed using flow cytometry. The histograms show mean of three experiments + SEM, with the exception of primary cell lines GBM03 (N = 1), and GBM04 (N = 2), where mean + SD are plotted. *p < 0.05, **p < 0.01.

As GBM cells are known to harbour mutations in a number of genes involved in cell cycle control (Catalogue of somatic mutation in cancer (COSMIC), http://cancer.sanger.ac.uk/cosmic), we verified that these cells were capable of undergoing cell cycle arrest by exposing them to 10 µM etoposide for 24 h. All three cell lines displayed a profound S phase arrest (Fig. S3). To investigate whether the observed cell cycle response was specific to GBM cells or common to other cancer cells, we exposed HeLa cells to 0.1%, 1% and 20% O2 for up to 72 h. As with the GBM cells, HeLa cell cycle distribution was unaffected by exposure to 1% O2. An increase in cells in G1 phase was observed following 48 h in 0.1% O2; however, this increase was not significant and cell cycle distribution returned to normal after 72 h, as previously observed in D566 cells (Fig. 3B).

The effect of hypoxia on the expression of HIF-1α and cell cycle regulatory proteins

We confirmed the ability of GBM cells to respond to hypoxia by measuring HIF-1α stabilisation using western blot (Fig. 4A). In all three cell lines, HIF-1α was stabilised in hypoxia, with increased protein levels in 0.1% compared to 1% O2 (Fig. 4B). To confirm that hypoxia does not have a significant effect on cell cycle regulation, mRNA levels of the cyclin-dependent kinase inhibitors p21, p27 and the well described transcription factor involved in G1/S transition, E2F1, were measured by qRT-PCR following up to 96 h in 0.1% O2 (Fig. 4C). No significant changes in p21, p27, or E2F1 expression were observed in any of the cell lines. D566 cells showed a non-significant increase in p21 expression at 24 h, with expression levels peaking at 48 h and decreasing after 72 h, consistent with the temporary G1 phase accumulation observed in D566 cells exposed to 0.1% O2 (Fig. 3A). Western blotting was used to investigate whether this transient increase in p21 mRNA was translated at the protein level. Only a small increase in protein was detected after 24 h exposure to 0.1% O2 (Fig. 4D). Furthermore, this increase was negligible compared to the amount of p21 detected following treatment with etoposide, a potent inducer of cell cycle arrest (see Fig. S3).

Figure 4 Hypoxia increases HIF-1α but does not change p21 protein expression.

Protein expression was examined by western blot following incubation in 20%, 1% or 0.1% O2 for 24, 48 or 72 h. Blots are representative examples taken from three experiments. (A) Western blots of HIF-1α protein expression. (B) Densitometry of HIF-1α protein expression shown in (A). (C) qRT-PCR was used to assess mRNA levels of p21, p27 and E2F1 following exposure to 0.1% O2. Values were normalised to cyclophilin A and expressed as fold change from 20% O2. Figure shows mean of three experiments + SEM. (D) Western blot of p21 protein expression. Cell exposed to the chemotherapeutic etoposide (Etop) were included as a control. The densitometry analysis, measured as the mean background subtracted integrated density for three experiments + SEM, has been plotted.

Discussion

Our results show that GBM cells survive and proliferate following chronic exposure to 1% O2, which is typical of the oxygen levels found in brain tumours. We have shown that only severe hypoxia (0.1% O2) is capable of inducing GBM cell cycle arrest and, furthermore, this arrest is only transient. Previous research has indicated that hypoxia causes cell cycle arrest; however, in the majority of these experiments cells were exposed to severe hypoxia or anoxia, which is not representative of the oxygen levels found within tumours (Box & Demetrick, 2004; Graeber et al., 1994; Amellem & Pettersen, 1991; Gardner et al., 2001). Although a change in cell cycle distribution has been reported in U87 cells exposed to 1% O2, no changes were observed in the growth curve (Li et al., 2013). Furthermore, as previous investigations have used short-term exposures (≤48 h) it is not clear whether the reported cell cycle changes are transient, as we have observed in D566 cells.

While none of the cell lines tested showed any changes to cell cycle distribution following exposure to pathophysiological levels of oxygen (1% O2) U251 cells displayed a reduction in cell number after 96 h. This suggests that in certain cell lines progression through each phase of the cell cycle may be slowed following long-term exposure to hypoxia. A G1 cell cycle arrest was only observed in one cell line (D566 cells) following exposure to severe hypoxia. This transient arrest was associated with a parallel increase in p21 mRNA; however, the increase was not significant and not translated at the protein level.

No changes in p27 or E2F1 gene expression were detected. Previous investigations have demonstrated an association between the cell cycle arrest induced by severe hypoxia and the expression of p21 and p27, although there is controversy as to whether these proteins are essential (Gardner et al., 2001; Goda et al., 2003; Graff et al., 2005). Goda et al. (2003) reported that HIF-1α mediated induction of p21 and p27 is essential for hypoxia-induced G1 cell cycle arrest. Gardner et al. (2001) also reported that hypoxia induces p27; however, in contrast to Goda et al. they found no involvement of HIF-1α in hypoxia-induced G1 arrest. The reason for these discrepancies is unclear but may be due to differences in cell lines and methods.

When we compared cell proliferation under physiological oxygen levels (8% O2, ‘physioxia’) to standard cell culture conditions (20% O2), no changes were detected. These findings contrast with previous research which has suggested a pro-proliferative role for physioxia. Fibroblasts display an elevated growth rate and a delayed onset of replicative senescence when cultured in 3% O2. These effects are a result of the reduction in oxidative damage accumulated under low oxygen conditions (Chen et al., 1995; Parrinello et al., 2003). Although HIF-1α is only present in small amounts above 1% O2, increasing evidence suggests that it plays a pro-proliferative role in physiological oxygen conditions. Neural progenitor cells display increased proliferation in 10% O2 in an effect mediated by HIF-1α, and the moderate induction of HIF-1α in 5% O2 has been shown to promote proliferation of both cancer and non-cancer cell lines, suggesting an important role in normal cell physiology (Carrera et al., 2014; Zhao et al., 2008).

After 48 h exposure to severe hypoxia (0.1% O2), we found that U251 and U87 cells underwent cell death. This observation is in line with the well-established finding that cell survival in vitro is compromised by exposure to extreme hypoxia (Hall, Bedford & Oliver, 1966). In GBM, the most severe regions of hypoxia (as indicated by high EF5 binding) are adjacent to regions of necrosis (Evans et al., 2004). One of the defining characteristics of GBM is pseudopalisading necrosis: an area of necrosis that often arises following a vaso-occlusive event, surrounded by a dense arrangement of cells organised into parallel rows (Rong et al., 2006). The surviving cells on the periphery express high levels of HIF-1α and matrix metalloproteinases and are thought to be actively migrating outward towards a more favourable environment (Sondergaard et al., 2002; Zagzag et al., 2006; Brat et al., 2004). As such, regions of necrosis are associated with higher grade tumours and a poorer prognosis (Hammoud et al., 1996).

One mechanism by which hypoxia is thought to increase resistance to chemotherapy is the induction of cell cycle arrest. As traditional chemotherapeutics target rapidly diving cells, this is thought to allow hypoxic cells to escape cell death (Harrison & Blackwell, 2004). Our research suggests that only a small proportion of cells will be exposed to hypoxia severe enough to cause cell cycle arrest, and of these cells a proportion will undergo cell death. It is therefore likely that other mechanisms play a more important role in hypoxia-mediated chemoresistance, such as the increased expression of the ABCB1 and p-glycoprotein drug transporters (Chou et al., 2012). Tumour hypoxia is also associated with increased expression of O6-methylguanine-DNA-methyltransferase (MGMT), a DNA repair protein involved in resistance to temozolomide (Pistollato et al., 2010). Decreased diffusion of drugs to hypoxic areas due to an increase in distance from blood vessels is also likely to play a major role (Tredan et al., 2007).

In conclusion, our results indicate that GBM cell proliferation and survival is not affected by pathophysiological levels of hypoxia, and that even severe hypoxia has only minimal effects on cell cycle. As such, the proliferation of the majority of cells in high-grade gliomas will not be affected by the reduced oxygen supply that is characteristic of these tumours. These findings illustrate the importance of using physiologically relevant oxygen concentrations before drawing conclusions about the effects of hypoxia on cell fate.

Supplemental Information

Table S1 Parameters used for qPCR

Click here for additional data file.

Figure S1 Severe hypoxia causes a reduction in U87 cell viability

U87 cells were exposed to 0.1% and 20% O2 for the indicated time points. Cell survival was assessed by Annexin/PI staining and flow cytometry.

Click here for additional data file.

Figure S2 Severe hypoxia causes a transient G1 phase arrest in D566 cells

Representative histograms of D566 cell cycle following exposure to 0.1% and 20% O2 for the indicated time points. Cell cycle distribution was analysed using flow cytometry.

Click here for additional data file.

Figure S3 Cell cycle arrest can be induced in GBM cells

Cells were incubated in 20% O2 and exposed to 10 µM etoposide for 24 h. Cell cycle distribution was analysed using flow cytometry.

Click here for additional data file.

We thank Dr. Carol Walker for coordinating access to patient tumour samples through the Walton Research Tissue Bank. We also extend our thanks to Mr Andrew Brodbelt of the Walton Centre NHS Foundation Trust for obtaining the samples.

Additional Information and Declarations

Competing Interests

Author Contributions

Human Ethics

Data Availability

Violaine See is an Academic Editor for PeerJ.

Rosalie Richards conceived and designed the experiments, performed the experiments, analyzed the data, wrote the paper, prepared figures and/or tables, reviewed drafts of the paper.

Michael D. Jenkinson and Brian J. Haylock contributed reagents/materials/analysis tools, reviewed drafts of the paper.

Violaine See conceived and designed the experiments, wrote the paper, reviewed drafts of the paper.

The following information was supplied relating to ethical approvals (i.e., approving body and any reference numbers):

The Walton Research Tissue Bank, Walton Centre NHS Foundation Trust. National Research Ethics Service (11/WNo03/2).

The following information was supplied regarding data availability:

figshare; https://figshare.com/s/6bfd585c89dd5c321f03.

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
