# Peer review of "Cell cycle progression in glioblastoma cells is unaffected by pathophysiological levels of hypoxia"

_PeerJ, doi:10.7717/peerj.1755_

## Round 0.1 · original submission · Minor Revisions

Both reviewers feel that the manuscript addresses interesting issue, however I ask you to address the minor issues that they raised.

Reviewer 1 ·

Basic reporting

Richards et al investigated the effect of different oxygen levels on glioblastoma
(GBM) cell proliferation and survival.
Results indicated that cell cycle distribution and viability are unaffected by long term exposure (up to 4 days) to pathophysiological levels of oxygen (1-8% O 2). Both transient cell cycle arrest and small amounts of cell death could only be detected when cells were exposed to severe hypoxia (0.1% O2). No significant changes in p21 protein expression levels were detected.

Th overall presentation of the data and structure of the manuscripts (quality of the figures, text clarity, equilibrium of different parts of the manuscript) are satisfying.

Experimental design

Experimental results are generally sound and clearly illustrated.

Minor points.
The decrease in viability at 96h of D566 (Fig 1A) does not look statistically significant and as such should not be emphasized.
U87 cells look unaffected by hypoxia for 72 hours, not 96 (see fig 2A and line 153 of the text).

Validity of the findings

Findings are of interest. Their impact would be stronger if mostly based on human GBM neurospheres that mirror the actual biology of GBM better than cells grown in serum.
Conclusions are coherent with results.

Additional comments

Li et al (Int J Med Sci. 2013; 10(4): 399–407) have also shown similar data on two of the cell lines used in this study, U87 and U251, adding data on their increased “stemness” under hypoxia. It would be fair to add that reference and specific comments, if any.

Reviewer 2 ·

Basic reporting

No comments

Experimental design

By using three different glioblastoma cancer cell lines (D566, U251 and U87) and two patients-derived cell lines (GBM3 and GBM04), the authors demonstrate that hypoxia does not affect cell proliferation (evaluated by cell counting with flow citometry) and cell death (evaluated by Annexin V/PI staining). Cells seem to respond only at prolonged exposure to severe hypoxia.

Validity of the findings

Although the results seem to be preliminar, the manuscript is clear and well written and the addressed issue is interesting. This reviewer has only minor revisions to report.
MINOR REVISIONS:
• The manuscript should be carefully checked for possible errors. Some mistakes are present:
- Abstract, line 22: “the effect of a range of oxygen tensions….were assessed” – please check.
- Introduction, Line 44: “characterised hystopathology” – please check.
- Results, line 170: “hypoxia-induced”.
- Discussion, line 215: repetition of “that”.
- Discussion, line 223: “hypoxia-induced”.
• Please check and correct “O 2” in all the figure legends.
• Please check all the histograms. All the x axis are missing.

---

## Round 0.2 · accepted · Accept

You have completely satisfied all reviewers criticism.